# Unveiling the Neem (*Azadirachta indica*) Effects on Biofilm Formation of Food-Borne Bacteria and the Potential Mechanism Using a Molecular Docking Approach

**DOI:** 10.3390/plants13182669

**Published:** 2024-09-23

**Authors:** Ghada Abd-Elmonsef Mahmoud, Nahed M. Rashed, Sherif M. El-Ganainy, Shimaa H. Salem

**Affiliations:** 1Botany and Microbiology Department, Faculty of Science, Assiut University, Assiut 71516, Egypt; shimaa.hassan@aun.edu.eg; 2Department of Arid Land Agriculture, College of Agricultural and Food Sciences, King Faisal University, P.O. Box 420, Al-Ahsa 31982, Saudi Arabia; salganainy@kfu.edu.sa; 3Horticulture Department, Faculty of Agriculture, Damietta University, Damietta 34519, Egypt; 4Plant Pathology Research Institute, Agricultural Research Center, Giza 12619, Egypt

**Keywords:** neem, antibacterial, human pathogens, phytochemicals, leaves, natural products

## Abstract

Biofilms currently represent the most prevalent bacterial lifestyle, enabling them to resist environmental stress and antibacterial drugs. Natural antibacterial agents could be a safe solution for controlling bacterial biofilms in food industries without affecting human health and environmental safety. A methanolic extract of *Azadirachta indica* (neem) leaves was prepared and analyzed using gas chromatography–mass spectrometry for the identification of its phytochemical constituents. Four food-borne bacterial pathogens (*Bacillus cereus*, *Novosphingobium aromaticivorans*, *Klebsiella pneumoniae*, and *Serratia marcescens*) were tested for biofilm formation qualitatively and quantitatively. The antibacterial and antibiofilm properties of the extract were estimated using liquid cultures and a microtiter plate assay. The biofilm inhibition mechanisms were investigated using a light microscope and molecular docking technique. The methanolic extract contained 45 identified compounds, including fatty acids, ester, phenols, flavonoids, terpenes, steroids, and antioxidants with antimicrobial, anticancer, and anti-inflammatory properties. Substantial antibacterial activity in relation to the extract was recorded, especially at 100 μg/mL against *K. pneumoniae* and *S. marcescens*. The extract inhibited biofilm formation at 100 μg/mL by 83.83% (*S. marcescens*), 73.12% (*K. pneumoniae*), and 54.4% (*N. aromaticivorans*). The results indicate efficient biofilm formation by the Gram-negative bacteria *S. marcescens*, *K. pneumoniae*, and *N. aromaticivorans*, giving 0.74, 0.292, and 0.219 OD at 595 nm, respectively, while *B. cereus* was found to have a low biofilm formation potential, i.e., 0.14 OD at 595 nm. The light microscope technique shows the antibiofilm activities with the biofilm almost disappearing at 75 μg/mL and 100 μg/mL concentrations. This antibiofilm property was attributed to DNA gyrase inhibition as illustrated by the molecular docking approach.

## 1. Introduction

Recently, unexpected increases in human disease incidence and disease outbreaks related to food-borne pathogens have been documented [1]. Pathogenic bacteria in food result in about one million deaths annually all over the world [2]. In the United States, food-borne pathogens cause 48 million illnesses and 3000 deaths each year with economic losses of over 78 billion dollars, and, from 2011 to 2017, caused 842 outbreaks and 14.237 illnesses each year [3]. The four essential mechanisms for successful food contamination include bacterial adherence, attachment, colonization, and survival of the pathogen [4]. There is substantial evidence that pathogenic bacteria adhering and developing speedily in high numbers in the form of a biofilm represent the main cause of human food diseases, and about 60% of human infections result from biofilm formation [5]. 

Bacterial biofilm represents one of the significant pathogenic strategies established by disease-causing bacteria, since it enables them to resist disinfection, antibiotics desiccation, and UV radiation, as well as the natural immune system of our bodies [6]. These bacterial barrier shields stand aggressively against the natural host defense mechanisms and external antibiotic treatments. By forming biofilm, about 1% of normal bacteria could transform into persisters, which tolerate lethal antibiotic concentrations and become more resistant to therapy [7]. Also, biofilm formation is responsible for 70% of microbial-inducing infections [8]. Bacterial biofilms that are found as a part of the natural environment are classified as neutral biofilms, while those developed on infection wounds are classified as harmful biofilms. Bacterial biofilms cause severe damage in terms of human life, health, industry concerns (contaminated industrial products), and the economy (equipment degradation and infection control) [8]. Moisturized surfaces that provide nutrients are recommended as ideal biofilm development environments. *Bacillus cereus*, *Escherichia coli*, *Klebsiella*, *Salmonella*, and *Staphylococcus* species can form biofilms on most surface materials, foods, and injuries under various environmental conditions [1,9].

Alternative natural medicine currently represents the primary type of care in 80% of developing countries [10]. It has been found that plant extracts have efficient inhibiting abilities against bacterial biofilms, rendering them suitable for chemical treatments [11]. *Senegalia nigrescens*, *Hypericum connatum*, *Eugenia erythrophylla*, *Syzygium legatii*, *Berginia ciliata*, *Piper betle*, *Ficus exasperate*, and *Searsia penduline* alcoholic extracts have been utilized as biofilm inhibitors against *Chromobacterium violaceum*, *Campylobacter coli*, *C. jejuni*, *Bacillus cereus*, *Enterococcus faecalis*, *Staphylococcus aureus*, *E. coli*, *Pseudomonas aeruginosa*, and *Salmonella typhimurium* [12,13,14,15,16,17,18]. *Azadirachta indica* (neem) belongs to the Meliaceae family that is utilized in traditional medicine as a human disease treatment [19]. *Azadirachta indica* (A. Juss) was first published in 1830 in Mém. Mus. Hist. Nat. 19: 221 (https://powo.science.kew.org/taxon/urn:lsid:ipni.org:names:1213180-2#publications, accessed on 9 August 2024). It originally came from India and Myanmar and was named the village pharmacy due to its numerous health properties such as anti-inflammation, diabetic control, anticancer, and even insect-repellent abilities [20]. Neem leaf extracts are rich in a wide range of bioactive compounds with various pharmacological potentials, including terpenes, flavonoids tannins, sterols, alkaloids, reducing sugar, and phenols which have fungicidal, antihistamine, antipyretic, anti-inflammatory, antibacterial, and antiseptic properties [20,21,22,23]. In this study, we explored the use of a methanolic extract from neem leaves as a novel, feasible, and cost-effective antibiofilm and antibacterial agent. We identified the bioactive compounds in the extract using gas chromatography–mass spectrometry (GC–MS). Gas chromatography–mass spectrometry is a very important technique when studying plant extracts, as it has been utilized to identify the different phytoconstituents found in the extracts, identifying their structures and also clearing their area compared with other constituents [24].

Neem’s unique antimicrobial phytoconstituents have resulted in their involvement in food safety and medicinal fields. The antibacterial activity of leaf extracts has been demonstrated against *E. coli*, *Staphylococcus*, *Streptococcus*, and *Pseudomonas* species [25]. Several reports have discussed the effectiveness of natural plant extracts as antibiofilm agents, which can be more effective than the common chemically synthesized antibiotic treatments, with lower or no side effects. The methanolic extract of *A. indica* leaves has been tested on fifty-eight multidrug-resistant bacteria of clinical origin (*Staphylococcus aureus*, *S. epidermidis, Pseudomonas aeruginosa*, and *S. saprophyticus*), and it has been found that the extract inhibits the biofilms with 80–85% success rate [26]. The antibacterial ability of phytochemical extracts has been known for ages, but recently, research has become more focused on the mechanisms of these extracts as antimicrobial agents. By identifying the extract constituents and testing their antimicrobial activities, we could explain the antimicrobial activities through the computational technique called molecular docking. Molecular docking involves a computational technique which depends on detecting the binding affinity of ligands (like plant extract components) to receptor proteins (like microbial pathogens). We applied a molecular docking approach to assess how these compounds inhibit DNA gyrase, which is an essential enzyme for bacterial growth. Recently, several studies have utilized this method to understand how antibacterial agents interact with their target proteins, and this technique is also used in bioactive research to develop the research into drug design or development [27]. The most common ones stated that the phytochemicals target the bacterial membrane, inhibit bacterial biofilm formation, and suppress the bacterial virulence factors (enzymes and toxins), [28].

Although plant extracts are known to provide an undenied basis for novel drug development, the antibiofilm activities are still not completely explored [29]. This study aimed to prepare methanolic leaves extract of neem (*Azadirachta indica*) and identify their bioactive constituents. Then, we tested the antibacterial activities of the methanolic extract against four human pathogenic bacteria recovered from contaminated food (*Bacillus cereus*, *Novosphingobium aromaticivorans*, *Klebsiella pneumoniae*, and *Serratia marcescens*), evaluating the biofilm activities of the bacterial strains and testing the abilities of the neem extract to control the biofilm formation. In doing so, we illustrate the possible biofilm controlling mechanisms depending on the bioactive compounds in the neem methanolic extract using the computational technique of molecular docking.

## 2. Results

### 2.1. Phytochemical Analysis of Neem (A. indica) Using Gas Chromatography–Mass Spectrometry Analysis

The GC–MS analysis of the methanol leaf extracts of *A. indica* recorded peaks corresponding to the bioactive compounds that were recognized by relating their peak retention time, peak area (%), height (%), and mass spectral fragmentation patterns to the known compounds described by the National Institute of Standards and Technology (NIST) library. Results revealed that 44 compounds were identified in *A. indica* leaf extracts (Figure 1 and Appendix A). The D-Glucose, 4-O-a-D-glucopyranosyl was the most recognizable compound with a peak area percentage of 13.03, which was followed by the diterpene alcohol phytol (8.31), 5-(1-Ethoxy-ethoxy)-4-methyl-hex-2-enal (6.73), and heptasiloxane-tetradecamethyl (4.32). Cycloheptasiloxane, tetradecamethyl and pseudosolasodine diacetate were found at peak areas 4.4 and 3.32, respectively. Phenols 2,2′-methylenebis[6-(1,1-dimethylethyl)-4-methy, benzofuran, 2,3-dihydro, 2-methoxy-4-vinylphenol, and 2,4-dimethoxyphenol were recorded as phenolic compounds, while sarreroside and geranyl isovalerate were found as flavonoids. Fatty acid esters (linoleic acid ethyl ester, 5,8,11,14-eicosatetraenoic acid, methyl ester, and hexadecanoic acid, methyl ester), fatty acids with polyphenols (ricinoleic acid) and flavonoids (erucic acid, and oleic acid) were also recorded. Ascorbic acid 2,6-dihexadecanoate was recorded as an antioxidant compound, ergosta-5,22-dien-3-ol, acetate (3a,22E) as a steroid compound, and dihydroxy-2,5-dimethyl-3(2H)-furan-3-one as an aroma compound. 1-Monolinoleoylglycerol trimethylsilyl ether, estra-1,3,5(10)-trien-17a-ol, cyclooctasiloxane, hexadecamethyl, heptasiloxane, hexadecamethyl, 5a-pregn-16-en-20-one, 3a,12a-dihydroxy-, diacetate, and cholestan-3-ol, 2-methylene (3a,5a) were all recorded in the extract, and all of these compounds were documented as antimicrobial, anticancer, and anti-inflammatory compounds.

### 2.2. Qualitative and Quantitative Assessment of Biofilm Formation

After cultivating the four bacterial isolates *B. cereus*, *N. aromaticivorans*, *K. pneumoniae*, and *S. marcescens* on modified Congo Red Agar medium for 72 h, the biofilm-forming isolates turned into a black color (*N. aromaticivorans*, *K. pneumoniae*, and *S. marcescens*), while in non-biofilm forming isolates like *B. cereus*, the color did not change, as shown in Figure 2. For quantitative estimation of the biofilm, a microtiter plate assay was used. The OD measurements at 595 nm indicated the biofilm formed significantly (*p* < 0.05), the highest biofilm formation was in *S. marcescens* (Sm-26) giving 0.74 ± 0.12 OD at 595 nm, which was followed by *K. pneumoniae* (Kp-38) 0.292 ± 0.04 OD at 595 nm. However, *N. aromaticivorans* (ASU 35) recorded 0.219 ± 0.06 OD at 595 nm, and *B. cereus* (ASU 36) showed a low ability for biofilm formation, 0.14 ± 0.007 OD at 595, which was in agreement with the qualitative data shown in Figure 3. The three bacterial isolates with high biofilm formation, *S. marcescens* (Sm-26), *K. pneumoniae* (Kp-38), and *N. aromaticivorans* (ASU 35), were selected for further experiments.

### 2.3. Antibacterial Properties of Neem (A. indica) Methanolic Extract

The antibacterial activities of neem (*A. indica*) methanolic extract were performed against *N. aromaticivorans*, *K. pneumoniae*, and *S. marcescens* in liquid cultures. The extracts showed high antibacterial properties against the three bacterial types with significant effects (*p* < 0.05) compared with the standard antibiotic agent (chloramphenicol) as shown in Figure 4a–c. By increasing the extract concentration, the antibacterial activities significantly (*p* < 0.05) increase, reaching the highest activities at 100 μg/mL. The highest antibacterial activities were recorded against *K. pneumonia* giving 1.2 ± 0.012, 0.796 ± 0.052, 0.696 ± 0.048, 0.624 ± 0.008, and 0.5 ± 0.036 × 10^8^ CFU/mL using 10, 25, 50, 75, and 100 μg/mL extract, respectively, compared with 3.284 ± 0.108 × 10^8^ CFU/mL for the non-treated (control) sample. However, chloramphenicol recorded 2.056 ± 0.016, 1.832 ± 0.032, 1.2 ± 0.32, 0.692 ± 0.02, and 0.52 ± 0.016 × 10^8^ CFU/mL for *K. pneumonia* using 10, 25, 50, 75, and 100 μg/mL extract, respectively (Figure 4a). *Serratia marcescens* came second and recorded 1.58 ± 0.052 (10 μg/mL), 1.36 ± 0.032 (25 μg/mL), 1.12 ± 0.02 (50 μg/mL), 1.03 ± 0.016 (75 μg/mL), and 0.896 ± 0.064 (100 μg/mL) × 10^8^ CFU/mL compared with 3.068 ± 0.22 × 10^8^ CFU/mL for the non-treated (control) sample. However, chloramphenicol recorded 2.012 ± 0.012, 1.804 ± 0.028, 1.57 ± 0.02, 1.48 ± 0.024, and 1.4 ± 0.028 × 10^8^ CFU/mL for *S. marcescens* using 10, 25, 50, 75, and 100 μg/mL extract, respectively (Figure 4b). *Novosphingobium aromaticivorans* was the least affected and recorded 1.67 ± 0.088 (10 μg/mL), 1.35 ± 0.032 (25 μg/mL), 1.3 ± 0.036 (50 μg/mL), 1.2 ± 0.024 (75 μg/mL), and 1.06 ± 0.048 (100 μg/mL) × 10^8^ CFU/mL compared with 3.044 ± 0.044 × 10^8^ CFU/mL for the non-treated (control) sample. Chloramphenicol was also affected against *N. aromaticivorans* and recorded 1.87 ± 0.016, 1.66 ± 0.028, 1.4 ± 0.016, 1.2 ± 0.008, and 1.09 ± 0.02 × 10^8^ CFU/mL for using 10, 25, 50, 75, and 100 μg/mL extract, respectively (Figure 4c).

### 2.4. Bacterial Biofilm Inhibition Using Neem (A. indica) Extract

A microtiter plate assay was used to estimate the bacterial biofilm inhibition of *N. aromaticivorans*, *K. pneumoniae*, and *S. marcescens* using five concentrations of the tested methanolic extract 10, 25, 50, 75, and 100 μg/mL, as shown in Figure 5, Figure 6, Figure 7 and Figure 8. The extract affects significantly (*p* < 0.05) the biofilm formation and recorded its highest results at 100 μg/mL. The biofilm formation of *S. marcescens* was affected by all of the extract concentrations and inhibited with 20.85% at 10 μg/mL, 34.84% at 25 μg/mL, 64.5% at 50 μg/mL, 82% at 75 μg/mL, and 83.83% at 100 μg/mL compared with 86.1% inhibition of gentamicin (50 μg/mL) using the positive control, as shown in Figure 5. These results match with the light microscopic photos which show the degradation of *S. marcescens* biofilm in the treated samples compared with the control sample (non-treated); the biofilm starts to disappear in the 50 μg/mL extract and almost disappeared in the 100 μg/mL extract, as shown in Figure 6. *Klebsiella pneumoniae* came second in the biofilm inhibition, and the extract recorded 25.7% at 10 μg/mL, 31.9% at 25 μg/mL, 55.8% at 50 μg/mL, 72.1% at 75 μg/mL, and 73.12% at 100 μg/mL *K. pneumoniae* biofilm inhibition compared with 74.5% inhibition of gentamicin (50 μg/mL) using the positive control, as shown in Figure 5. Light microscopic photos of *K. pneumoniae* biofilm showed the biofilm degradation in treated samples disappeared with the control sample (non-treated), the biofilm started to disappear in 75 μg/mL extract and almost disappeared in 100 μg/mL (Figure 7). For *N. aromaticivorans* biofilm inhibition and the extract recorded the lowest results compared with *K. pneumoniae*, and *S. marcescens* giving 13.1% at (10 μg/mL), 45.9% at (25 μg/mL), 51.5% at (50 μg/mL), 53.2% at (75 μg/mL), and 54.4% at (100 μg/mL) *N. aromaticivorans* biofilm inhibition, comparing with 59.5% inhibition of gentamicin (50 μg/mL) using the positive control as shown in Figure 5. Light microscopic photos of *N. aromaticivorans* biofilm showed the biofilm degradation in treated samples compared with the control sample (non-treated); the biofilm started to significantly degrade by increasing the extract concentration, especially at 75 and 100 μg/mL; however, the degradation process was lower than that for *K. pneumoniaew* and *S. marcescens* (Figure 8).

### 2.5. Antibiofilm and Antibacterial Mechanisms of Neem Extract Using Molecular Docking Approach

Six compounds (**4**, **10**, **14**, **17**, **18**, and **20**) showed a promising binding affinity with DNA gyrase, and all the binding energies are listed in Table 1. Compound **4** forms one hydrogen bond with ARG76 and exhibits a hydrophobic interactions with VAL120, ILE90, VAL167, and ILE78. Compound **10** forms three hydrogen bonds with ARG136, THR165, and ASP73 and shows a hydrophobic interaction with ILE78 Figure 9. Compound **14** forms a hydrogen bond with ASN46 and shows a hydrophobic interaction with VAL120, ILE78, and ILE90. Compound **17** exhibits a hydrophobic interaction with VAL43, VAL71, VAL120, ILE78, and ILE90, as shown in Figure 10. Compound **18** shows a hydrophobic interaction with VAL71, THR165, and ILE78. Compound **20** forms a hydrophobic interaction with VAL120, VAL167, ILE78, and THR165, as shown in Figure 11. To validate our docking results, we performed an RMSD calculation and superimposed the redocked ligand onto the co-crystallized ligand. The RMSD, measured by aligning the redocked complex with the co-crystallized complex, was 1.792 Å, which is within the success range, as shown in Figure 12.

## 3. Discussion

Natural products have traditionally been utilized as antioxidant, antimicrobial, and preservative agents to protect food from spoilage [30]. The ethanol extract of *Senegalia nigrescens* and *Hypericum connatum* was an efficient antibiofilm of *Chromobacterium violaceum* [12] and *Pseudomonas aeruginosa* [13], respectively. *Eugenia erythrophylla*, *E. umtamvunensis* and *Syzygium legatii* leaf extracts have inhibited the biofilm formation of *B. cereus*, *E. faecalis*, *S. aureus*, *E. coli*, *P. aeruginosa*, and *S. typhimurium* [14]. *Berginia ciliata* rhizome extract inhibited *Pseudomonas aeruginosa* biofilm by 80% [15]. An ethanolic extract of *Piper betle* inhibited the biofilm of *Staphylococcus aureus* and *Escherichia coli* [16]. *Ficus exasperata* extract showed good antibiofilm properties against *E. coli*, *Campylobacter coli*, and *C. jejuni* [17]. The biofilm of *Bacillus cereus* was susceptible to *Searsia pendulina* extract with 98.22% [18]. Damaging the bacterial cell wall represents the major antibiofilm mechanism of natural compounds [31], which is followed by peptidoglycan synthesis inhibition [32]. Also, flavonoids in plant extracts were found to inhibit the bacterial biofilm via quorum sensing (QS) inhibition, which works on inhibiting the initiation of the biofilm [33]. Terpenoids in plant extracts could modify the bacterial cell membrane (the fatty acid composition), which causes a hydrophobicity of the bacterial cells, disrupting the biofilm [34].

Forty-four compounds were found in the methanolic extracts including fatty acids, esters, hydrocarbons, alkaloids, antioxidants, flavonoids, terpenes, and phenols. A single plant extract can consist of hundreds or even thousands of bioactive compounds with varied structures named phytochemicals [35]. These secondary metabolites have significant healing properties that facilitate infectious treatment for a long time; their antimicrobial properties are related to the presence of alkaloids, glycosides, flavonoids, phenols, saponins, steroids, and terpenoids [36]. In our results, D-glucose, 4-O-a-D-glucopyranosyl and heptasiloxane1,1,3,3,5,5,7,7,9,9,11,11,13,13-tetradecamethyl were the higher peak area compounds in the neem methanolic extract; these compounds were documented as having significant antibacterial and antitumor roles [37,38]. In our extract, we detected glyceryl monolinoleate, which has been considered an antiviral agent, while ascorbic acid 2,6-dihexadecanoate was documented as an antioxidant compound with properties protecting against diseases like the common cold, wounds, and lowered skin infections [39], and it was also found as an efficient inhibitor of *Escherichia coli* biofilm [40]. Oleic acid and erucic acid in the extract were previously identified as flavonoids in the conjugated fatty acids with antimicrobial and antioxidant effects [41]. Cycloheptasiloxane, tetradecamethyl, and pseudosolasodine diacetate were all reported as antioxidant and anticancer compounds [42]. Eicosatetraenoic acid and methyl ester were reported as omega-3 fatty acids with biofilm inhibition activities [43]. Quinolinone was documented as an antibiofilm of *E. coli*, *P. aeruginosa*, and *C. neoformans* [44], while hexadecanoic acid controls the biofilm of *C. albicans*, *C. glabrata* and *C. tropicalis* [45].

Using neem (*A. indica*) methanolic extract showed high antibacterial activities against *S. marcescens*, *K. pneumoniae*, and *N. aromaticivorans*, reaching about 75% growth inhibition at 100 μg/mL concentration. Hikaambo et al. [46] stated that *Azadirachta indica* leaf extract has antibacterial activities against *Escherichia coli*. Altayb et al. [23] found that *Azadirachta indica* leaf extract has antibacterial activities against *P. aeruginosa*, *Citrobacter* spp., *K. pneumoniae*, and *E. coli* using 50% extract as the effective concentration and attributed this activity to fatty acids, Beta d-Mannofuranoside, and geranyl compounds as the dominant and most active ones. Ibrahim and Kebede [47] showed that *Salmonella typhi*, *Klebsiella pneumoniae*, *B. subtilis*, and *E. coli* growth were controlled significantly using the neem extract. However, Kaur et al. [48] found that neem extracts could control *Bacillus anthracis*, *E. coli*, and *S. aureus* efficiently. Ali et al. [24] found that *A. indica* extract has effective antibacterial activities against *Pasteurella multocida*, *Salmonella pullorum*, and *Escherichia coli* using 25 to 100 mg/mL of leaf extract. Morales-Covarrubias et al. [49] demonstrated that 62.5 mg/mL of neem extract was significant in inhibiting *Vibrio parahaemolyticus* in foods. Ravva and Korn [50] stated that neem leaf extract was able to inhibit *Escherichia coli* O157:H7 in foodstuffs. Moreover, glycolipids and glycerides neem leaves showed inhibitory activities to *Salmonella typhi*, *E. coli*, *Shigella dysenteriae*, and *Vibrio cholera* [51].

Recently, infectious diseases of bacterial pathogens have been the most global health issue that causes high mortality due to the rise of drug resistance and loss of the efficacy of conventional medicines [52]. The major food-borne pathogens form biofilms, which magnify human infections in humans and make them difficult to cure [11]. During our study, three Gram-negative bacteria, *Novosphingobium aromaticivorans* (ASU 35), *Klebsiella pneumoniae* (Kp-38), and *Serratia marcescens* (Sm-26), and one Gram-positive bacteria *Bacillus cereus* (ASU 36) recovered from fresh foods were tested for biofilm formation, and it was found that Gram-negative bacteria formed biofilms more efficiently than Gram-positive ones. This finding is in agreement with several researchers who found that the common food-borne biofilm-forming bacteria are mostly from the Gram-negative types including *Serratia* spp., *Klebsiella* spp. *Clostridium* spp., *Salmonella enterica*, *Escherichia coli*, *Pseudomonas* spp., and a few Gram-positive bacteria like *Listeria monocytogenes*, *Bacillus* spp., and *Staphylococcus* spp., [53]. Moreover, Gram-negative bacteria are becoming a critical global threat due to their nosocomial status and multidrug resistance, especially *Pseudomonas*, *Klebsiella*, *Acinetobacter*, *Serratia*, *Escherichia*, and *Enterobacter* genera [54]. From previous bacterial types, *Klebsiella pneumoniae*, *Streptococcus pneumoniae*, *Escherichia coli*, *Staphylococcus aureus*, and *Pseudomonas aeruginosa* are reported as the five most dangerous biofilm-forming bacteria responsible for 54.9% of human deaths; this biofilm formation causes rapid cell and tissue damage [8].

In our results, *Serratia marcescens* and *Klebsiella pneumoniae* recorded high efficiencies in the biofilm formation, especially *S. marcescens*. *Serratia marcescens* is a pathogenic bacteria related to the family Enterobacteriaceae that causes food spoilage and spread in all food types as well as water bodies using biofilm formation [55]. It causes respiratory infections, wound infections, meningitis, and septicaemia [56]. It used a quorum-sensing mechanism for the biofilm formation as well as the production of carbapenem, prodigiosin. and enzyme-related virulence [57]. Meanwhile, *Klebsiella pneumoniae* causes pneumonia, wound infections, bloodstream infections, and meningitis, and it represents the most significant opportunistic bacteria [58]. *Klebsiella pneumoniae* is widely distributed in all food sources and possesses a capsule (a polysaccharide) which helps it in pathogenesis by preventing phagocytosis [59]. The capsule lipopolysaccharides also contribute to the formation of the biofilm [58]. *Bacillus cereus* is also a major food-poisoning bacteria that forms biofilms and resistant endospores that enable it to tolerate harsh environmental conditions [60]. *Bacillus cereus* is recorded as an opportunistic pathogen present in foods like dairy products, breakfast cereals, meat, spices, chicken, fruits, vegetables, and sweets [61]. Diarrheal, fever, bacteremia, non-gastrointestinal illness, endocarditis, respiratory illness, and central nervous system illness are all the results of its infection [62,63]. Diarrhea, fever, systematic infection, and vomiting are all the symptoms of bacterial biofilm infections [64].

By using neem (*A. indica*) methanolic extract, biofilm inhibition percentage increased by increasing the extract concentration, reaching 83.8% for *S. marcescens*, 73.1% for *K. pneumoniae*, and 54.4% for *N. aromaticivorans* biofilm inhibition at 100 μg/mL concentration. Natural plant substances provide high efficiency in controlling the *Klebsiella pneumoniae* biofilm and inhibit cell growth, including eugenol and *Andrographis paniculata* [65]. This could be attributed to the low electron density generated inside the periplasmic space and the cytoplasm that creates cell fluid discharge. Lahiri et al. [66] stated that the phenolic and flavonoids in neem extract demonstrated the highest biofilm degradation in dental plaque, which consisted mainly of *Porphyromonas gingivalis* and *Alcaligenes faecalis* and also caused a reduction in quorum sensing and the genetic material of the biofilm-generating cells. Guchhait et al. [67] found that neem extract had efficient antibiofilm activity against *Staphylococcus. aureus* and *Vibrio cholerae* using 100 to 300 μg/mL concentrations. Neem (*A. indica*) methanolic extract inhibited the dental biofilm of human teeth containing *E. faecalis*, *S. mutans*, and *S. aureus* [68]. Quelemes et al. [69] found that neem leaf extract inhibits methicillin-resistant *S. aureus* biofilm using 125 μg/mL extract. In human studies, neem-based toothpaste reduces the bacterial biofilm of *Staphylococcus mutans* in the mouth and teeth plaque [70]. Wylie and Merrell [19] found that the neem leaf extract controls the methicillin-resistant bacterial biofilm of *S. aureus.* Neem leaf extract has also been found as an effective biofilm inhibitor of *Streptococcus viridans*, *Staphylococcus aureus*, *Porphyromonas gingivalis* human pathogenic bacterial [71], and *Helicobacter pylori* biofilm [72].

We used molecular docking to investigate the molecular mechanisms behind the significant biological activities of the compounds found in the methanolic extract *of Azadirachta indica*. These compounds have shown strong antibacterial and antibiofilm effects, which may stem from their ability to inhibit bacterial DNA gyrase. Six compounds have the abilities to bind bacterial DNA gyrase, which represents essential enzymes in the bacterial formations and catalyzes the ATP for coiling the DNA double-stranded for closing the DNA circular [73]. This enzyme is crucial for bacterial growth because it plays a key role in DNA replication, transcription, and supercoiling [74]. Inhibiting DNA gyrase ultimately leads to bacterial cell death. The impressive antimicrobial and antibiofilm properties of these compounds are likely due to their unique structural characteristics, which include various aromatic groups and polar functionalities. These structures enable the compounds to engage in a range of non-covalent interactions, such as hydrophobic interactions and hydrogen bonding, with the active site of DNA gyrase. The RMSD value indicated a close similarity between the conformation of the redocked ligand and that of the experimental ligand in the enzyme’s active site, thus validating the docking results. Overall, neem leaves extract’s potential in combating bacterial biofilms opens up exciting possibilities for new treatments and management strategies in various fields.

## 4. Materials and Methods

### 4.1. Collection and Extraction of Neem (A. indica) Leaves

Fresh leaves of the neem (*Azadirachta indica*) tree were collected from Mansoura University, Egypt in December 2022 in clean and dry plastic bags. The collected leaves were washed and rinsed to remove dust and other impurities. They were then air-dried under a shade for 15 days; then, a total of 50 g of leaves were crushed using a mortar and pestle, and 80% methanol was used to soak the leaves for three days with daily filtration and evaporation. Then, by using a rotary evaporator apparatus under reduced pressure, the solvent was evaporated to dryness. The extracts were then covered and stored in a refrigerator for further use [75].

### 4.2. Phytochemical Analysis of Neem (A. indica) Extract

The gas chromatography–mass spectrometry method was used to conduct a qualitative and quantitative characterization of neem extract, using the model (GC–MS-QP2010-Ultra) from Shimadzu Company, Kyoto, Japan, with a capillary column Rtx-5MS column (30 m, 0.25 mm, 0.25 μm). The split mode was used for sample injection, and it was operated in electron ionization (EI) mode at 70 eV with an inflow rate of 1.69 mL/min. Helium gas was used as the carrier gas. The injector temperature was set at 300 °C, the temperature of the ion source was 200 °C, and 250 °C was used as the interface temperature. The oven temperature program was as follows: the initial temperature at 50 °C rising at 7 °C/min to 180 °C; then, the rate changed to 10 °C/min, reaching the final temperature at 280 °C with 2 min as the hold time. In a total 22-minute run, the sample was analyzed by the scan mode in a range of 40 to 500 m/z charges to ratio.

### 4.3. Bacterial Culture, Growth, and Maintenance

Four bacterial isolates were recovered from fresh foods, including fresh sushi (*Salmo salar*) and rabbitfish (*Siganus rivulatus*). The isolates include one Gram-positive bacteria, *Bacillus cereus* (ASU 36), and three Gram-negative bacteria: *Novosphingobium aromaticivorans* (ASU 35), *Klebsiella pneumoniae* (Kp-38), and *Serratia marcescens* (Sm-26) [76,77]. The bacterial isolates were stored in 20% glycerol at −8 °C, and before the experiment, the bacterial cultures were re-grown on nutrient agar (NA) plates for 24 h at 35 °C. After growth, a single bacterial colony from each isolate was transferred in nutrient broth (1% peptone, 1% beef extract, and 0.6% NaCl) for 24 h in a shaking incubator at 220 rpm at 35 °C (Lab-Line 3597 Orbital Environmental Shaker, CA, USA) for inoculums preparations. After growth, bacterial cells were collected, centrifuged at 5000× *g* for 15 min, washed, and suspended in water saline with 10^7^ CFU/mL concentration [78].

### 4.4. Assessment of the Biofilm Formation Capabilities

The four bacterial isolates *B. cereus*, *N. aromaticivorans*, *K. pneumoniae*, and *S. marcescens* were tested for biofilm formation on modified Congo Red agar medium containing sucrose, 30; beef extract, 5; peptone, 5; NaCl, 3; Congo Red, 0.8; and agar, 20 g/L. A single colony that was 24 h old from each isolate was streaked on the medium surfaces and incubated at 37 ± 1 °C for 72 h. After incubation, the biofilm formation was observed by turning the bacterial colonies into a black color, while the non-biofilm bacterial producers maintained their color [79].

### 4.5. Quantitative Determination of Bacterial Biofilm

Biofilm production by *B. cereus*, *N. aromaticivorans*, *K. pneumoniae*, and *S. marcescens* was estimated using the microtiter plate assay following Cruz et al. [80]. A single colony from each bacterial isolate was inoculated into Luria–Bertani medium (yeast extract, 0.5%; tryptone, 1.5%; NaCl, 0.05%; agar, 2%) supplemented with sucrose (2%) and incubated overnight at 37 °C in a rotary shaker (150 rpm). A 96-well microplate with a flat bottom was used; each well was filled with 200 μL of 24 h bacterial growth suspension, and other wells contained free-cell medium as a negative control. Then, the plates were incubated at 37 °C for 48 h, and after incubation, the well contents were carefully discarded and washed with 200 μL of phosphate buffer (pH 7.3) three times to discard the non-adherent cells. Wells with adhered biofilm were fixed with methanol and air-dried. Crystal violet (2%) was utilized to stain the biofilm; then, 33% glacial acetic acid was added to resolubilize the adhered biofilm. Finally, the optical density (OD) of stained bacterial biofilm was detected at 595 nm, indicating the biofilm formation using the microtiter plate reader (BioTek EPOCH, Winooski, VT, USA).

### 4.6. Testing the Antibacterial Properties of Neem (A. indica) Extract

The antibacterial activities of neem (*A. indica*) methanolic extract were performed against the three pathogenic bacteria *N. aromaticivorans*, *K. pneumoniae*, and *S. marcescens* recovered from contaminated fresh foods. The antibacterial activities were carried out in 10 mL sterilized nutrient broth containing five concentrations of the plant extract (10, 25, 50, 75, and 100 μg/mL), 100 µg/mL chloramphenicol as a positive control, and medium-free antibacterial agents as a negative control. The tubes were inoculated with 1% bacteria inoculums and incubated at 37 °C for 48 h. The OD was estimated at 595 nm and the bacterial colony-forming unit (CFU/mL) was calculated.

### 4.7. Assessment of Bacterial Biofilm Inhibition Using Neem (A. indica) Extract

The antibiofilm activity of *A. indica* methanolic extract against bacterial biofilm formation by the Gram-negative bacteria *N. aromaticivorans*, *K. pneumoniae*, and *S. marcescens* was assessed using the microtiter plate assay following Yimgang et al. [81]. A 96-well microplate with a flat bottom was used; each well was filled with 100 μL of overnight bacterial culture and 10 μL of the extract at five concentrations (10, 25, 50, 75, and 100 μg/mL) against the positive control (PC) gentamicin (50 μg/mL) and incubated for 48 h at 37 °C. After incubation, free-floating cells were removed, and the bacterial biofilms were washed with phosphate buffer (pH 7.3). Then, the biofilms were stained with crystal violet 2%, and the absorbance values were measured using the microtiter plate reader at 595 nm (BioTek EPOCH, Winooski, VT, USA). Wells containing medium only were considered as a negative control, and gentamicin was used as a positive control. The percentage of biofilm inhibition was calculated as follows:% of inhibition = [(control OD_595nm_ − treated OD_595nm_)/control OD_595nm_] × 100.

### 4.8. Light Microscopy Analysis of the Biofilm Formation

Sterilized falcon tubes (50 mL) containing sterilized glass pieces (1 × 1 cm) were prepared, and then 10 mL of sterilized LB medium growth media was added. Four concentrations (10, 25, 50, 75, and 100 μg/mL) of the methanolic extract were prepared and injected into tubes with bacterial inoculum (1%) added. The tubes were incubated for 24 h at 37 °C. Then, the glass slides were collated, washed with double-distilled water, and stained with 0.4% crystal violet. The stained-glass slides were examined for biofilm formation using a light microscope (Olympus CX41, Tokyo, Japan).

### 4.9. Molecular Docking

Autodock vina 1.5.6 [82] was used to perform the molecular docking of *Azadirachta indica* methanolic extract constituents against DNA gyrase. The complex crystal structure of DNA gyrase with its inhibitor (Clorobiocin) was retrieved from the Protein Data Bank with PDB ID: 1KZN [74]. All ligands and water molecules were removed, and the PDBQT files were prepared accordingly. The average binding energies of docking was used to evaluate the binding affinity with the enzyme active site. The ligand–protein complexes showing the lowest binding energies have been chosen and visually analyzed using PyMOL software (version 2.5.4). The grid size parameters were x = 42, y = 48, z = 44.

### 4.10. Statistical Analysis

The statistical analysis for the data was analyzed using Statistix 8.1 software (one-way ANOVA) with *p* value < 0.05, and all the data are presented as the mean ± standard deviations.

## 5. Conclusions

By increasing the pathogen’s antibiotic resistance each day, natural antimicrobial agents of plant origin are promising antibiotic alternatives in food and medical fields. One particular advantage is their efficient way of degrading the bacterial biofilms, which is a serious issue and represents the first step in food contamination and human disease initiation. Neem methanolic extract contains 44 bioactive metabolites including terpenes, phenols, flavonoids, antioxidants, reducing sugar, sterols and fatty acid esters, which are documented as antimicrobial, anticancer, and anti-inflammatory agents. These compounds proved high antibacterial properties and efficiencies in controlling the bacterial biofilms with 54.4 to 83.83% at 100 μg/mL, which was demonstrated clearly in the light microscopic photos. Using the molecular docking technique, we illustrated the potential mechanism of these bioactive compounds in degrading the bacterial biofilm. Six compounds can bind to bacterial DNA gyrase, which represents an essential enzyme in bacterial formations by catalyzing the ATP for coiling the double-stranded DNA and closing the circular DNA.

In future studies, we aim to optimize the production process of these bioactive compounds and define the best plant conditions for production with more in-depth studies of the phytochemicals mechanisms for preventing microbial diseases. We need to combine these phytochemicals with other antibacterial agents in ratios and test their antimicrobial activities in single and combined treatments. In addition, we will also conduct pre-clinical studies to determine their cytotoxicity and the effective dosages for future applications in clinical trials.

## Figures and Tables

**Figure 1 plants-13-02669-f001:**
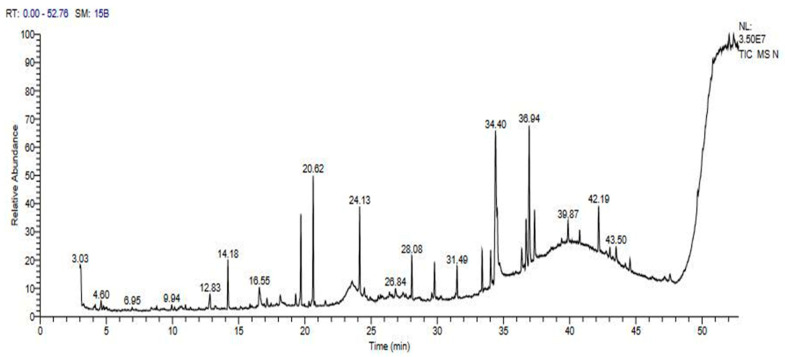
Gas chromatography–mass spectrometry (GC-MS) chromatogram of the methanolic extract of neem (*A. indica*); numbers above the peaks represented the retention times.

**Figure 2 plants-13-02669-f002:**
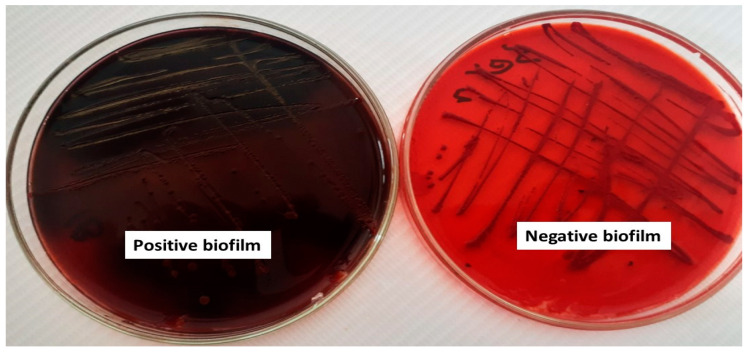
Qualitative assessment of bacterial biofilm using Congo Red agar medium showing the black color as positive biofilm formation and no color changes as negative biofilm formation.

**Figure 3 plants-13-02669-f003:**
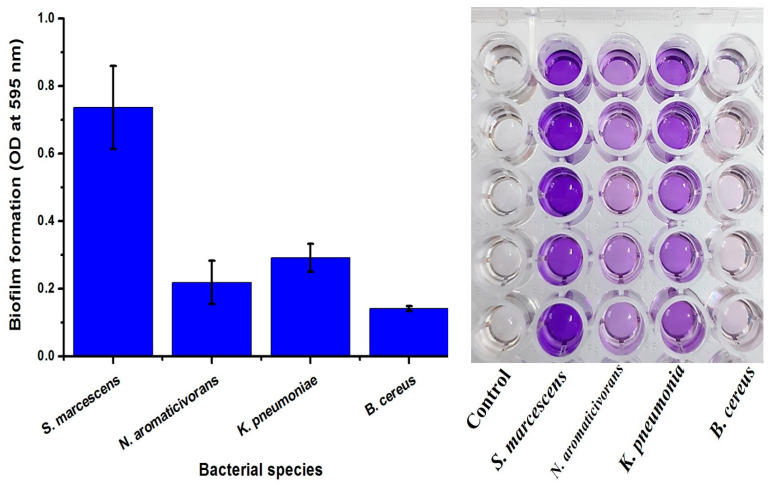
Biofilm biomass assessment of *B. cereus*, *N. aromaticivorans*, *K. pneumoniae*, and *S. marcescens* (OD at 595 nm) using microtiter plate assay.

**Figure 4 plants-13-02669-f004:**
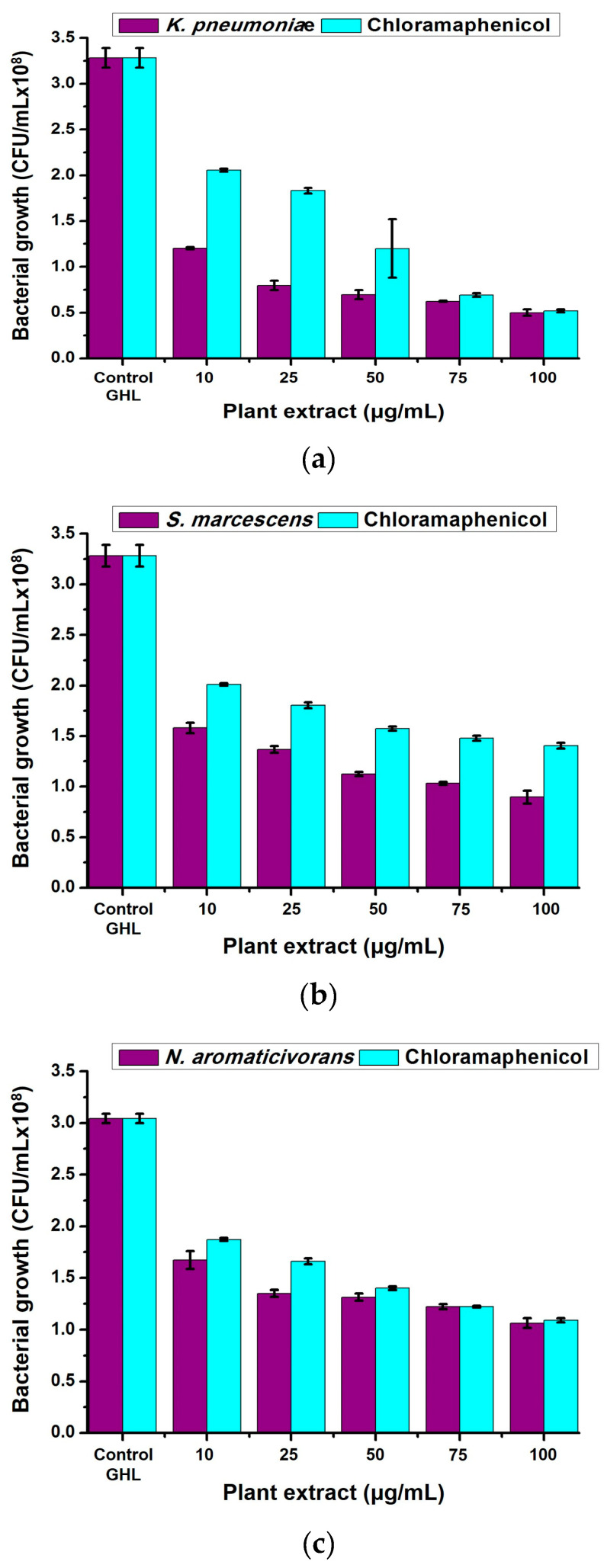
Antimicrobial activities of neem (*A. indica*) extract (10, 25, 50, 75, and 100 μg/mL) against (**a**): *K. pneumoniae*, (**b**): *S. marcescens*, and (**c**): *N. aromaticivorans* bacterial growth comparing with chloramphenicol (CHL) as positive control.

**Figure 5 plants-13-02669-f005:**
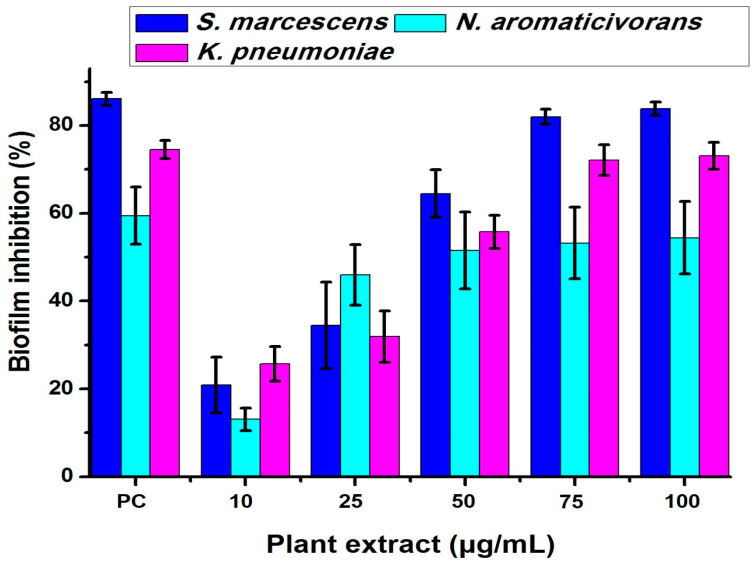
Antibiofilm efficacy of different concentrations of neem (*A. indica*) extract on *S. marcescens*, *K. pneumoniae*, and *N. aromaticivorans* as assessed by crystal violet quantification of biofilm. Gentamicin was used as the positive control (PC).

**Figure 6 plants-13-02669-f006:**
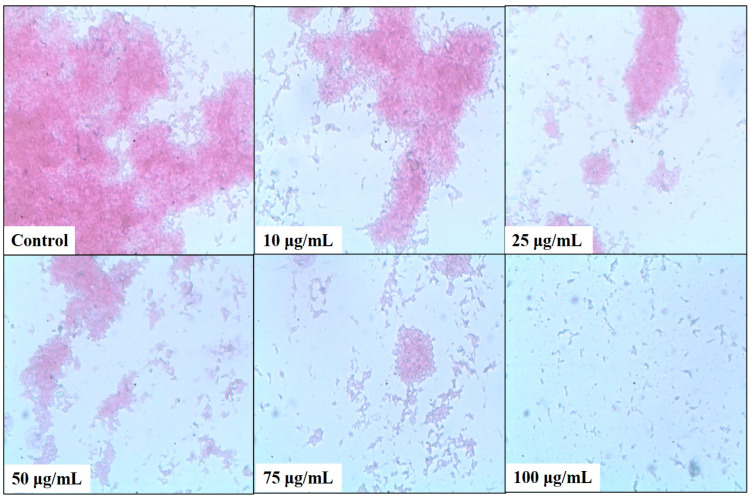
Light microscopic analysis of *Serratia marcescens* (Sm-26) biofilm treated with 10, 25, 50, 75, and 100 μg/mL of *A. indica* methanolic extract compared with the control sample.

**Figure 7 plants-13-02669-f007:**
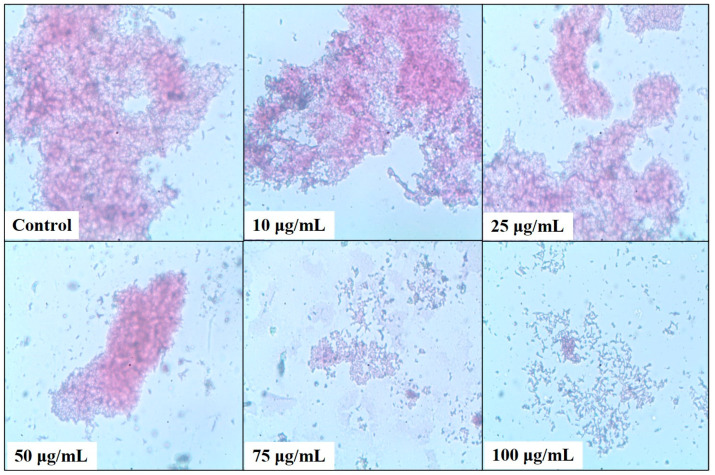
Light microscopic analysis of *Klebsiella pneumonia* (Kp-38) treated with 10, 25, 50, 75, and 100 μg/mL of *A. indica* methanolic extract compared with the control sample.

**Figure 8 plants-13-02669-f008:**
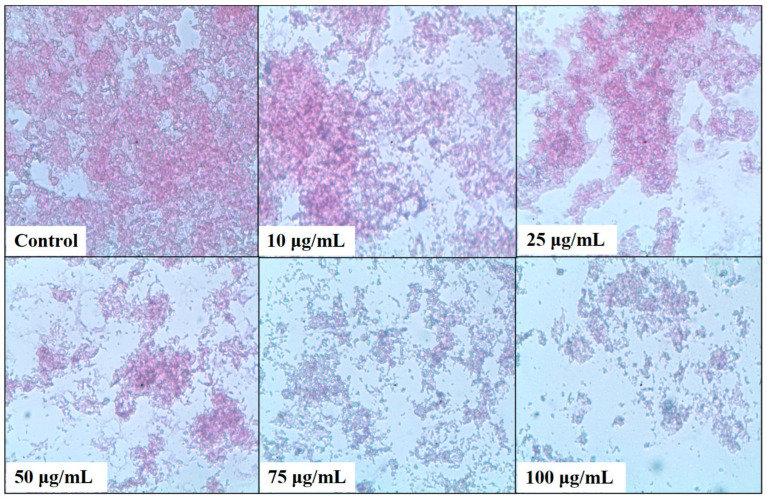
Light microscopic analysis of *Novosphingobium aromaticivorans* (ASU 35) treated with 10, 25, 50, 75, and 100 μg/mL of *A. indica* methanolic extract compared with the control sample.

**Figure 9 plants-13-02669-f009:**
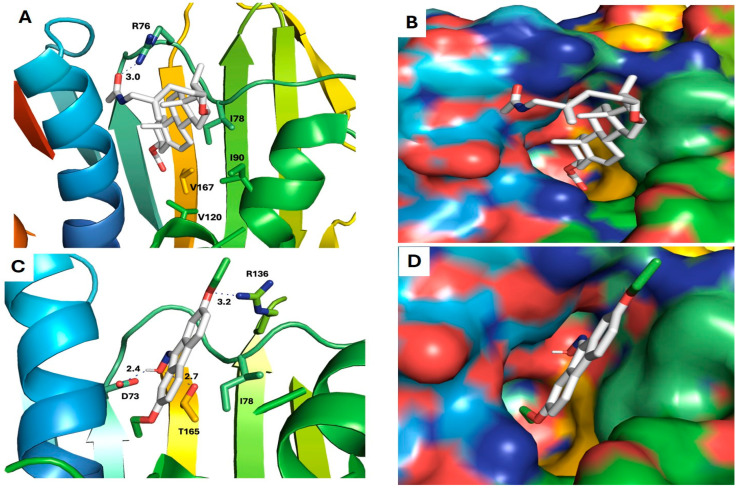
Docking models of pseudosolasodine diacetate and 9-oximino-2,7-diethoxyfluorene with DNA gyrase. (**A**,**B**) Stick model and surface map of pseudosolasodine diacetate docked model with DNA gyrase; it forms a hydrogen bond with ARG76. (**C**,**D**) Stick model and surface map of 9-oximino-2,7-diethoxyfluorene docked model with DNA gyrase; it forms hydrogen bonds with ARG136, THR165, and ASP73.

**Figure 10 plants-13-02669-f010:**
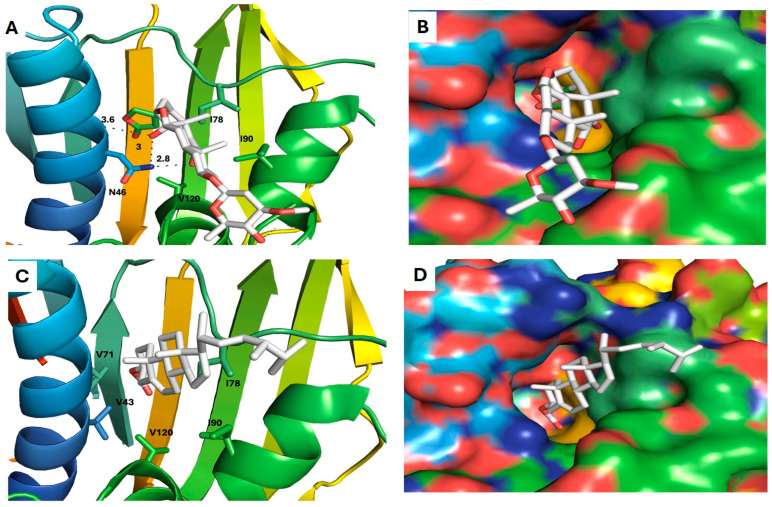
Docking models of sarreroside and ergosta-5,22-dien-3-ol, acetate, (3a,22E)- with DNA gyrase. (**A**,**B**) Stick model and surface map of sarreroside docked model with DNA gyrase; it forms a hydrogen bond with ASN46. (**C**,**D**) Stick model and surface map of Ergosta-5,22-dien-3-ol, acetate, (3a,22E)-docked model with DNA gyrase; it exhibits hydrophobic interactions with VAL43, VAL71, VAL120, ILE78, and ILE90.

**Figure 11 plants-13-02669-f011:**
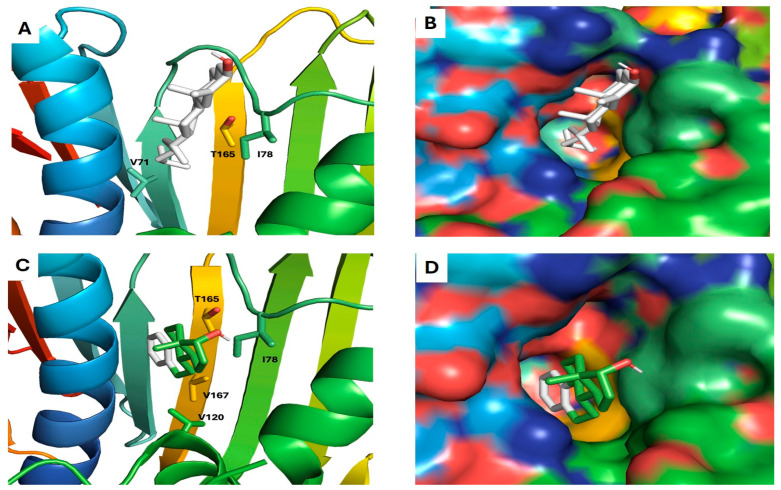
Docking models of cholestan-3-ol, 2-methylene-, (3a,5a)- and estra-1,3,5(10)-trien-17a-ol with DNA gyrase. (**A**,**B**) Stick model and surface map of cholestan-3-ol, 2-methylene-, (3a,5a)- docked model with DNA gyrase; it exhibits hydrophobic interactions with VAL71, THR165, and ILE78. (**C**,**D**) Stick model and surface map of estra-1,3,5(10)-trien-17a-ol docked model with DNA gyrase; it exhibits hydrophobic interaction with VAL120, VAL167, ILE78, and THR165.

**Figure 12 plants-13-02669-f012:**
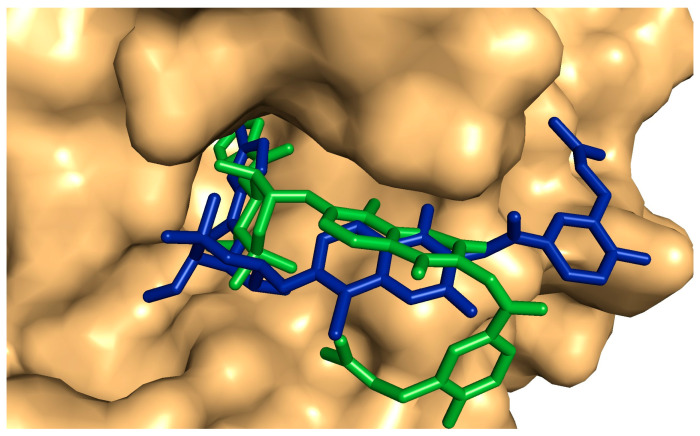
Surface representation of the superimposed redocked protein–ligand complex on the co-crystalized complex. Redocked ligand is blue, co-crystalized ligand is green.

**Table 1 plants-13-02669-t001:** Calculated binding energies of *Azadirachta indica* methanolic extract constituents and Clorobiocin with 1KZN.

No.	Compounds	Calculated Binding Energies (kcal/mol)	Peak Area (%)
1	(2-Methyl-[1,3]dioxolan-2-yl)-acetic acid, phenyl ester	−5.68 ± 0.50	1.42
2	2-Acetonyl-9-[3-deoxy-a-d-ribouranosyl]hypoxanthine	−6.34 ± 0.44	0.21
3	Ethyl iso-allocholate	−6.67 ± 0.38	0.85
4	Pseudosolasodine diacetate	−7.07 ± 0.45	3.32
5	4-Methoxy-6-methyl-5-nitroisobenzofuran-1,3-dione	−5.82 ± 0.33	0.12
6	Benzofuran, 7-(2,4-dinitrophenoxy)-3-ethoxy-2,3-dihydro-2,2-dimethyl-	−6.2 ± 0.33	0.03
7	Methyl 4-O-acetyl-2,3,6-tri-O-ethyl-a-d-mannopyranoside	−4.54 ± 0.23	0.58
8	2-[4-methyl-6-(2,6,6-trimethylcyclohex-1-enyl)hexa-1,3,5-trienyl]cyclohex-1-en 1 carboxaldehyde	−6.6 ± 0.35	0.80
9	Preg-4-en-3-one, 17a-hydroxy-17a-cyano-	−6.35 ± 0.34	0.06
10	9-Oximino-2,7-diethoxyfluorene	−7.12 ± 0.61	0.06
11	17a-Allyl-3a-acetoxy-17a-aza-D-homoandrost-5-ene-17-one	−6.95 ± 0.79	0.06
12	N-[3-Diethylaminopropyl]-4-oxo-1,2,3,4,5,6,7,8-octohydroquinoline	−4.8 ± 0.44	0.21
13	Carbonic acid, (ethyl)(1,2,4-triazol-1-ylmethyl) diester	−4.37 ± 0.32	0.16
14	Sarreroside	−7.53 ± 0.29	0.18
15	9,10-Secocholesta-5,7,10(19)-triene-3,24,25-triol, (3a,5Z,7E)-	−6.67 ± 0.22	0.24
16	Pyrimidine-2,4(1H,3H)-dione,1,3-dimethyl-6-[2-(4-morpholyl)ethenyl]-5-nitro-	−5.74 ± 0.36	0.11
17	Ergosta-5,22-dien-3-ol, acetate, (3a,22E)-	−7.1 ± 0.60	0.65
18	Cholestan−3-ol, 2-methylene-, (3a,5a)-	−7.35 ± 0.50	0.29
19	Pregn-4-ene-3,20-dione, 17,21-dihydroxy-, bis(O-methyloxime)	−6.11 ± 0.35	0.39
20	Estra-1,3,5(10)-trien-17a-ol	−7.00 ± 0.92	2.17
21	Clorobiocin (Standard)	−8.35 ± 0.68	
1KZN: The complex crystal structure of DNA gyrase with its inhibitor (Clorobiocin)			

## Data Availability

The data presented in this study are available upon request from the corresponding author.

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
