# Peer review of "Unveiling the Neem (Azadirachta indica) Effects on Biofilm Formation of Food-Borne Bacteria and the Potential Mechanism Using a Molecular Docking Approach"

_plants, 2024, doi:10.3390/plants13182669_

Round 1

Reviewer 1 Report

Comments and Suggestions for Authors

Ghada Abd-Elmonsef Mahmoud et al. Unveiling the neem (Azadirachta indica) effects on biofilm for mation of food-borne bacteria and the potential mechanism using molecular docking approach.

However, the manuscript still needs some minor revision before it can be accepted for publication. Please refer to my opinions below.

1.     In the third paragraph of introduction, the author can add the introduction that plants can inhibit biofilm formation.

2.     In the fourth paragraph of introduction, the author can add a brief introduction to the research methods of GC-MS and molecular docking.

3.     The standard antimicrobial agent (Chloramphenicol) can be indicated under the control group in figure 2.

4.     In the first paragraph of the discussion, we can quote the past research and compare the similarities and differences between the results of this study and the existing literature more specifically.

5.     It is very important to add some other effective plant compound extracts to the discussion part. The logic of the manuscript will be that it would be better if these plant compound extracts can also become effective biofilm inhibitors.

6.     The results of molecular docking can be discussed in more detail. Azadirachtin and Azadirachtin show strong binding affinity to the active sites of bacterial biofilm-related proteins, which can play a good role in destroying biofilm.

7.     The future research direction can be written at the conclusion. For example, the future research can explore the synergistic effect of Azadirachta indica extract with other natural antibacterial agents, and further clarify the molecular pathway of its anti-biofilm activity.

Author Response

Ghada Abd-Elmonsef Mahmoud et al. Unveiling the neem (Azadirachta indica) effects on biofilm for mation of food-borne bacteria and the potential mechanism using molecular docking approach.

However, the manuscript still needs some minor revision before it can be accepted for publication. Please refer to my opinions below.

Response: Many thanks for your recommendation. All your valuable comments have been addressed as below:

  1. In the third paragraph of introduction, the author can add the introduction that plants can inhibit biofilm formation.

Response: Thank you for this remark. The third paragraph of introduction have been modified as recommended, please see lines 61-68.

  1. In the fourth paragraph of introduction, the author can add a brief introduction to the research methods of GC-MS and molecular docking.

Response: Thank you for this remark. The fourth paragraph of introduction have been modified as recommended, please see lines 77-83, 94-102.

  1. The standard antimicrobial agent (Chloramphenicol) can be indicated under the control group in figure 2.

Response: Thank you for this remark. The figure has been corrected and chloramphenicol has been added.

  1. In the first paragraph of the discussion, we can quote the past research and compare the similarities and differences between the results of this study and the existing literature more specifically.

Response: Thank you for this remark. The paragraph has been modified, please see lines 281-289.

  1. It is very important to add some other effective plant compound extracts to the discussion part. The logic of the manuscript will be that it would be better if these plant compound extracts can also become effective biofilm inhibitors.

Response: Thank you for this remark. I have added this information; please see lines 302-314.

  1. The results of molecular docking can be discussed in more detail. Azadirachtin and Azadirachtin show strong binding affinity to the active sites of bacterial biofilm-related proteins, which can play a good role in destroying biofilm.

Response: Thank you for this remark. We discussed the molecular docking results in more detail., please see lines 390-406.

  1. The future research direction can be written at the conclusion. For example, the future research can explore the synergistic effect of Azadirachta indica extract with other natural antibacterial agents, and further clarify the molecular pathway of its anti-biofilm activity.

Response: Thank you for this remark. The conclusion part has been modified and future studies is added, please see lines 509-527.

Reviewer 2 Report

Comments and Suggestions for Authors

Azadirachta indica, also known as Nim or Neem, is a tree belonging to the Meliaceae family, native to India and Burma, used for centuries for its medicinal properties. Recent research has confirmed its anti-inflammatory, antiseptic and antiseborrheic actions, however, there are still no sufficient studies regarding the internal use of Neem, even if it already seems evident that this plant has considerable potential. In this work, the methanolic extract of A. indica leaves was analyzed using GC-MS for the identification of its phytochemical constituents. The antibacterial and anti-biofilm properties of the extract were estimated on four foodborne bacterial pathogens (B. cereus, N. aromaticivorans, K. pneumoniae and S. marcescens) using liquid cultures and the microtiter plate test. The biofilm inhibition mechanisms were illustrated using an optical microscope and a molecular docking technique.

The topic is interesting and falls within the aims and scope of the Journal. Overall, the structure of the manuscript is well set and the experimentation conducted allows to add new data to the previously published literature.

However, some changes are requested, as reported below.

In detail

Please revise the abstract to clarify some sentences (e.g. lines 14-16).

Lines 17-18: Amend the sentence:

Methanolic extract of A. indica leaves was prepared and analyzed using GC- Mass for identification of its phytochemical constituents.

with

Methanolic extract of Azadirachta indica (neem) leaves was prepared and analyzed using Gas chromatography mass spectrometry (GC-MS) for identification of its phytochemical constituents.

Please change the keywords to terms not present in the title.

Line 129: Change the sentence:

Gas chromatography–mass (GC-MS) spectrometry chromatogram…

with

Gas chromatography–mass spectrometry (GC-MS) chromatogram…

Paragraph 2.3 contains the explanation of why the three bacterial isolates were selected for further experiments. Therefore, it should be reported before paragraph 2.2.

Line 234: Please add a note below the table to explain the acronym 1KZN.

Line 298-302: Please change the sentence:

During our study, three Gram Novosphingobium aromaticivorans (ASU 35), Klebsiella pneumoniae (Kp-38), and Serratia marcescens (Sm-26) and one Gram-positive bacteria Bacillus cereus (ASU 36) recovered from fresh foods were tested for biofilm formation and it was found that Gram-negative bacteria formed biofilms efficiently than Gram-positive ones.

with

During our study, three Gram-negative bacteria Novosphingobium aromaticivorans (ASU 35), Klebsiella pneumoniae (Kp-38), and Serratia marcescens (Sm-26) and one Gram-positive bacteria Bacillus cereus (ASU 36) recovered from fresh foods were tested for biofilm formation and it was found that Gram-negative bacteria formed biofilms efficiently than Gram-positive ones.

The Conclusion section should be modified to be less generic and more related to the results obtained. It should also report the limitations of the study and possible future approaches.

Please add a list of abbreviations used in the text.

Comments on the Quality of English Language

The text is written in fluent English.

Author Response

Azadirachta indica, also known as Nim or Neem, is a tree belonging to the Meliaceae family, native to India and Burma, used for centuries for its medicinal properties. Recent research has confirmed its anti-inflammatory, antiseptic and antiseborrheic actions, however, there are still no sufficient studies regarding the internal use of Neem, even if it already seems evident that this plant has considerable potential. In this work, the methanolic extract of A. indica leaves was analyzed using GC-MS for the identification of its phytochemical constituents. The antibacterial and anti-biofilm properties of the extract were estimated on four foodborne bacterial pathogens (B. cereus, N. aromaticivorans, K. pneumoniae and S. marcescens) using liquid cultures and the microtiter plate test. The biofilm inhibition mechanisms were illustrated using an optical microscope and a molecular docking technique.

The topic is interesting and falls within the aims and scope of the Journal. Overall, the structure of the manuscript is well set and the experimentation conducted allows to add new data to the previously published literature. However, some changes are requested, as reported below.

Response: Many thanks for your recommendation. All your valuable comments have been addressed as below:

Please revise the abstract to clarify some sentences (e.g. lines 14-16).

Response: Thank you for this remark. The sentences have been modified.

Lines 17-18: Amend the sentence: Methanolic extract of A. indica leaves was prepared and analyzed using GC- Mass for identification of its phytochemical constituents.

with

Methanolic extract of Azadirachta indica (neem) leaves was prepared and analyzed using Gas chromatography mass spectrometry (GC-MS) for identification of its phytochemical constituents.

Response: Thank you for this remark. The sentence has been modified.

Please change the keywords to terms not present in the title.

Response: Thank you for this remark. The keywords have been modified to nim; antibacterial; human pathogens; phytochemicals; leaves; natural products.

Line 129: Change the sentence: Gas chromatography–mass (GC-MS) spectrometry chromatogram…

with

Gas chromatography–mass spectrometry (GC-MS) chromatogram…

Response: Thank you for this remark. The sentence has been modified.

Paragraph 2.3 contains the explanation of why the three bacterial isolates were selected for further experiments. Therefore, it should be reported before paragraph 2.2.

Response: Thank you for this remark. We transferred the 2.3. paragraph before 2.2. as recommended.

Line 234: Please add a note below the table to explain the acronym 1KZN.

 Response: Thank you for this remark. We added the explanation of the acronym 1KZN below Table 1. (1KZN: The complex crystal structure of DNA gyrase with its inhibitor (Clorobiocin)

Line 298-302: Please change the sentence: During our study, three Gram Novosphingobium aromaticivorans (ASU 35), Klebsiella pneumoniae (Kp-38), and Serratia marcescens (Sm-26) and one Gram-positive bacteria Bacillus cereus (ASU 36) recovered from fresh foods were tested for biofilm formation and it was found that Gram-negative bacteria formed biofilms efficiently than Gram-positive ones.

 with

 During our study, three Gram-negative bacteria Novosphingobium aromaticivorans (ASU 35), Klebsiella pneumoniae (Kp-38), and Serratia marcescens (Sm-26) and one Gram-positive bacteria Bacillus cereus (ASU 36) recovered from fresh foods were tested for biofilm formation and it was found that Gram-negative bacteria formed biofilms efficiently than Gram-positive ones.

 Response: Thank you for this remark. The sentence has been modified.

The Conclusion section should be modified to be less generic and more related to the results obtained. It should also report the limitations of the study and possible future approaches.

Response: Thank you for this remark. The conclusion part has been modified and future studies is added, please see lines 509-527.

Please add a list of abbreviations used in the text.

 Response: Thank you for this remark. All the abbreviations checked and were identified in the first time of mention it.

Comments on the Quality of English Language. The text is written in fluent English.

Response: Many thanks for your recommendation.

Reviewer 3 Report

Comments and Suggestions for Authors

The manuscript needs a major revision.

Main notes:

1.     The last sentence of the abstract is too general and can be painlessly deleted.

2.     It is worth adding the term "leaves" to the list of keywords since the manuscript title does not use this word, although the leaves of the plant were used to make the extract.

3.     In the introduction, it is worth indicating where the the studied plant grows as wild and as an introduced species (https://powo.science.kew.org/taxon/urn:lsid:ipni.org:names:1213180-2. It is also worth taking information from this site about the scientist who described this species for science during the first mention - Azadirachta indica A. Juss.

4.     When describing the relevance of the topic in the introduction, it is worth adding data about the importance of docking research in this field.

5.     It needs to improve the aim and rationale of the study at the end of the Introduction section. It is worth making this paragraph more concise. The repetition of the word "biofilm" in every sentence of this paragraph should be avoided: рр. 95-101.

6.     The conclusion section is too general and it does not contain any mention of neem tree extract. Therefore, it should be rewritten and improved.

7.     The abbreviations cannot be used in the Abstract without deciphering: line 17 – GC Mass? (Gas chromatography‑mass spectrometry). It is also worth checking whether this abbreviation is used correctly and uniformly throughout the text.

8.     The dimension ‘ml‘–should be written using the capital letter L everywhere – ‘mL’.

9.     Table S1 is missing in the text, and the accompanying file at the specified link (https://www.mdpi.com/xxx/s1) is missing.

10. A moderate check is required for the English language and style through the text.

11. And finally, analyzing the iThenticate report, it could be concluded that the authors should increase the percentage of originality of the text.

Comments on the Quality of English Language

A moderate check is required for the English language and style through the text.

Author Response

  1. The last sentence of the abstract is too general and can be painlessly deleted.

 Response: Thank you for this remark. The sentence has been deleted.

  1. It is worth adding the term "leaves" to the list of keywords since the manuscript title does not use this word, although the leaves of the plant were used to make the extract.

Response: Thank you for this remark. The keywords have been modified to nim; antibacterial; human pathogens; phytochemicals; leaves; natural products.

  1. In the introduction, it is worth indicating where the the studied plant grows as wild and as an introduced species (https://powo.science.kew.org/taxon/urn:lsid:ipni.org:names:1213180-2. It is also worth taking information from this site about the scientist who described this species for science during the first mention - Azadirachta indica A. Juss.

 Response: Thank you for this remark. The information’s have been added to the introduction, please see lines 69-80.

  1. When describing the relevance of the topic in the introduction, it is worth adding data about the importance of docking research in this field.

 Response: Thank you for this remark. The information’s have been added to the introduction, please see lines 94-102.

  1. It needs to improve the aim and rationale of the study at the end of the Introduction section. It is worth making this paragraph more concise. The repetition of the word "biofilm" in every sentence of this paragraph should be avoided: рр. 95-101.

 Response: Thank you for this remark. The paragraph has been modified, please see lines 107-115.

  1. The conclusion section is too general and it does not contain any mention of neem tree extract. Therefore, it should be rewritten and improved.

Response: Thank you for this remark. The conclusion part has been modified and future studies is added, please see lines 509-527.

  1. The abbreviations cannot be used in the Abstract without deciphering: line 17 – GC Mass? (Gas chromatography‑mass spectrometry). It is also worth checking whether this abbreviation is used correctly and uniformly throughout the text.

 Response: Thank you for this remark. The abbreviation has been deleted and checked throughout the whole manuscript.

  1. The dimension ‘ml‘–should be written using the capital letter L everywhere – ‘mL’.

Response: Thank you for this remark. I have corrected it and checked the whole manuscript and the figures.

  1. Table S1 is missing in the text, and the accompanying file at the specified link (https://www.mdpi.com/xxx/s1) is missing.

Response: Thank you for this remark. We are very sorry for this technical mistake; the table has been uploaded with the manuscript.

  1. A moderate check is required for the English language and style through the text.

 Response: Thank you for this remark. We have revised the English editing and grammar with a specific proofreading tool for academic publications and also by native speaker.

  1. And finally, analyzing the iThenticate report, it could be concluded that the authors should increase the percentage of originality of the text.

 Response: Thank you for this remark. We go through the manuscript and improved it for increase the percentage of originality.

Comments on the Quality of English Language

A moderate check is required for the English language and style through the text.

Response: Thank you for this remark. We have revised the English editing and grammar with a specific proofreading tool for academic publications and also by native speaker.

Reviewer 4 Report

Comments and Suggestions for Authors

The paper by Mahmoud and colleagues is focused on the effect of Azadirachta indica on biofilm. The topic is very interesting and addresses a current issue. Some improvements are required before further considering the paper.

The major concerns are related to the computational part:

-       Molecular docking: the authors must describe why DNA gyrase was selected as a target. Is it biologically relevant? How was the PDB file selected?

-       Table 1: “calculated binding energy” is more indicated than “binding energies”. Moreover, values should be indicated in kcal/mol (not Kcal/mol). Is the number a standard deviation? How was it calculated? This is rather unconventional. Moreover, please check the use of significant digits (e.g. compound 20).

-       How was the docking validated? Since a cognate ligand is present in the structure (1kzn) this could be used for re-docking and RMSD calculation.

- More experimental details are needed (number of runs, grid size)

Moreover, I report some minor general comments below:

-       In abstract, Azadirachta indica should be reported in extenso the first time.

-       Introduction is interesting, but I suggest to shorten the first part (until line 73).

-       Some formatting issues are present (check paragraph 2.1, the first line).

-       I suggest redesigning Figure 2. Some panels have different sizes (e.g., panel c is stretched).

-       Check the use of abbreviations, in particular in the case of scientific nomenclature.

Author Response

The paper by Mahmoud and colleagues is focused on the effect of Azadirachta indica on biofilm. The topic is very interesting and addresses a current issue. Some improvements are required before further considering the paper.

 Response: Many thanks for your recommendation. All your valuable comments have been addressed as below:

The major concerns are related to the computational part:

-       Molecular docking: the authors must describe why DNA gyrase was selected as a target. Is it biologically relevant? How was the PDB file selected?

Response: Thanks for this comment. We selected DNA gyrase as a target protein because this enzyme is essential for bacterial growth as it is involved in the replication, transcription, and supercoiling of bacterial circular DNA so, inhibiting DNA gyrase leads to bacterial death. We already described that in the manuscript (Discussion section, lines 390-406).

We have four different strains in our study and there is no available complex structure of DNA gyrase of these strains with known inhibitors. Therefore, we used the DNA gyrase of E. coli that exhibited high sequence homology with the DNA gyrase in our strains. In addition, the PDB used in our study is a complex structure with inhibitor at high resolution (2.3Å) so it is so useful to give more reliable docking data. Here is the sequence alignment with similarity score:

-       Table 1: “calculated binding energy” is more indicated than “binding energies”. Moreover, values should be indicated in kcal/mol (not Kcal/mol). Is the number a standard deviation? How was it calculated? This is rather unconventional. Moreover, please check the use of significant digits (e.g. compound 20).

Response: Thanks for this comment. As the reviewer suggested, we replaced “binding energies” with “calculated binding energies” in Table 1 legend. The values in Table 1 represent average of binding energies ± standard deviations. Upon completing the docking process, each compound exhibited nine models, and each model has its own binding energy. So calculated the average of the binding energy of the 9 models and calculated the standard deviation as well. We amended the significant digits in compound 20.

-       How was the docking validated? Since a cognate ligand is present in the structure (1kzn) this could be used for re-docking and RMSD calculation.

Response: Thanks for this comment. We removed the ligand from the PDB file and then we redocked again the ligand. The standard binding energy in Table 1 represents the re-docked value of the co-crystallized ligand. We included the RMSD values in the revised version of the manuscript and added Figure 12 to show the overlay between the redocked and co-crystallized ligand in the active site of DNA gyrase. 

- More experimental details are needed (number of runs, grid size)

Response: Thanks for this comment. We added the grid size parameters in the methods section.

Moreover, I report some minor general comments below:

 -       In abstract, Azadirachta indica should be reported in extenso the first time.

Response: Thank you for this remark. The sentence has been modified.

-       Introduction is interesting, but I suggest to shorten the first part (until line 73).

Response: Thank you for this remark. This paragraph has been modified as recommended.

-       Some formatting issues are present (check paragraph 2.1, the first line).

Response: Thank you for this remark. The formatting has been corrected.

-       I suggest redesigning Figure 2. Some panels have different sizes (e.g., panel c is stretched).

Response: Thank you for this remark. The figure size has been corrected.

-       Check the use of abbreviations, in particular in the case of scientific nomenclature.

Response: Thank you for this remark. All the scientific names were written in complete name when mentioned in the first time.

Round 2

Reviewer 3 Report

Comments and Suggestions for Authors

The authors have significantly improved the quality of the manuscript and it can now be recommended for publication

Author Response

The authors have significantly improved the quality of the manuscript and it can now be recommended for publication.

Thank you for your kind recommendation

Reviewer 4 Report

Comments and Suggestions for Authors

While the authors addresses most of the comments and motivated the modifications, a major concern remains about docking. After checking Fig.12, it can be noticed that the molecule has a bad superimposition with the cognate ligand. The re-docking demonstrates that the protocol is not fully suitable. The authors must address this point.

Author Response

While the authors addresses most of the comments and motivated the modifications, a major concern remains about docking. After checking Fig.12, it can be noticed that the molecule has a bad superimposition with the cognate ligand. The re-docking demonstrates that the protocol is not fully suitable. The authors must address this point.

Response: Thank you for your remark. We believe that the protocol used in our study is acceptable and produces trusted data. We calculated the RMSD, and it is 1.792 Å which lies in the acceptable range of the docking analysis according to the following reference:

  • Ramírez, D.; Caballero, J. Is It Reliable to Take the Molecular Docking Top Scoring Position as the Best Solution without Considering Available Structural Data? Molecules2018, 23, 1038. https://doi.org/10.3390/molecules23051038.

Figure 1. Examples of RMSD for docked ligands (gray) with respect to reference ligand at the crystal structures (green) for illustrating good (RMSD ≤ 2.0 Å), acceptable (RMSD > 2.0 Å and <3.0 Å) and bad (RMSD ≥ 3.0 Å) solutions in each target protein.

Please see also these 2 references:

  • Mena-Ulecia K, Tiznado W, Caballero J (2015). Study of the Differential Activity of Thrombin Inhibitors Using Docking, QSAR, Molecular Dynamics, and MM-GBSA. PLoS ONE 10(11): e0142774. https://doi.org/10.1371/journal.pone.0142774.

  • Gohlke, H., Hendlich, M., & Klebe, G. (2000). Knowledge-based scoring function to predict protein-ligand interactions. Journal of molecular biology, 295(2), 337-356. https://doi.org/10.1006/jmbi.1999.3371.

In Figure 12, the main backbone of the re-docked ligand is overlaid successfully with the co-crystallized ligand. This main backbone exhibits the main interaction pattern with the active site of the DNA gyrase as shown in the attached figures below. This emphasizes that our protocol is valid. However, there is a deviation in the branch chain orientation (which has no interaction with the active site of DNA gyrase). This deviation could be attributed to free rotation around the single bond which is very difficult to be predicted in docking. Therefore, it employed different orientations in docking analysis. In addition, the environment in the experimental model is different than the docking. For example, we have no water molecules in docking analysis which could affect the electrostatic interaction. All these variations could lead to this variation, which is always expected from docking studies compared to the experimental structural data.

Red ring in the figure represents the main backbone of the re-docked ligand (blue) with the co-crystallized ligand (green).

The highlighted aromatic moiety exhibited no interaction at all with the active site of DNA gyrase in the crystal structure PDB ID: 1KZN.

Round 3

Reviewer 4 Report

Comments and Suggestions for Authors

The authors motivated their response.

Author Response

Comments and Suggestions for Authors: The authors motivated their response.   Thank you for your kind recommendation